Log-based software monitoring: a systematic mapping study

http://orcid.org/0000-0003-0846-040X Cândido Jeanderson 1 2 j.candido@tudelft.nl
Aniche Maurício 1
http://orcid.org/0000-0003-4850-3312 van Deursen Arie 1
1 Department of Software Technology, Delft University of Technology , Delft , Netherlands
2 Adyen N.V. , Amsterdam , Netherlands
Huisman Marieke
Electronic publication date: 2021 May 6
Publication date: 2021
Volume: 7
Electronic Location ID: e489
Received 2020 Nov 20; Accepted 2021 Mar 22
Copyright: © 2021 Cândido et al.
Copyright year: 2021
Copyright holder: Cândido et al.
License: This is an open access article distributed under the terms of the Creative Commons Attribution License, which permits unrestricted use, distribution, reproduction and adaptation in any medium and for any purpose provided that it is properly attributed. For attribution, the original author(s), title, publication source (PeerJ Computer Science) and either DOI or URL of the article must be cited.
License URL: https://creativecommons.org/licenses/by/4.0/

Keywords: Logging practices, Log infrastructure, Log analysis, DevOps, Monitoring

Funding: Netherlands Organization for Scientific Research (NWO) MIPL 628.008.003 This work was supported by the Netherlands Organization for Scientific Research (NWO) MIPL project [grant number 628.008.003]. The funders had no role in study design, data collection and analysis, decision to publish, or preparation of the manuscript.

==============================
Modern software development and operations rely on monitoring to understand how systems behave in production. The data provided by application logs and runtime environment are essential to detect and diagnose undesired behavior and improve system reliability. However, despite the rich ecosystem around industry-ready log solutions, monitoring complex systems and getting insights from log data remains a challenge. Researchers and practitioners have been actively working to address several challenges related to logs, e.g., how to effectively provide better tooling support for logging decisions to developers, how to effectively process and store log data, and how to extract insights from log data. A holistic view of the research effort on logging practices and automated log analysis is key to provide directions and disseminate the state-of-the-art for technology transfer. In this paper, we study 108 papers (72 research track papers, 24 journals, and 12 industry track papers) from different communities (e.g., machine learning, software engineering, and systems) and structure the research field in light of the life-cycle of log data. Our analysis shows that (1) logging is challenging not only in open-source projects but also in industry, (2) machine learning is a promising approach to enable a contextual analysis of source code for log recommendation but further investigation is required to assess the usability of those tools in practice, (3) few studies approached efficient persistence of log data, and (4) there are open opportunities to analyze application logs and to evaluate state-of-the-art log analysis techniques in a DevOps context.

Introduction

Software systems are everywhere and play an important role in society and economy. Failures in those systems may harm entire businesses and cause unrecoverable loss in the worst case. For instance, in 2018, a supermarket chain in Australia remained closed nationwide for 3 h due to “minor IT problems” in their checkout system (Chung, 2018). More recently, in 2019, a misconfiguration and a bug in a data center management system caused a worldwide outage in the Google Cloud platform, affecting not only Google’s services, but also businesses that use their platform as a service, e.g., Shopify and Snapchat (Wired, 2019; Google, 2019).

While software testing plays an important role in preventing failures and assessing reliability, developers and operations teams rely on monitoring to understand how the system behaves in production. In fact, the symbiosis between development and operations resulted in a mix known as DevOps (Bass, Weber & Zhu, 2015; Dyck, Penners & Lichter, 2015; Roche, 2013), where both roles work in a continuous cycle. In addition, given the rich nature of data produced by large-scale systems in production and the popularization of machine learning, there is an increasingly trend to adopt artificial intelligence to automate operations. Gartner (2019) refers to this movement as AIOps and also highlights companies providing automated operations as a service. Unsurprisingly, the demand to analyze operations data fostered the creation of a multi-million dollar business (TechCrunch, 2017, Investor’s Business Daily, 2018) and plethora of open-source and commercial tools to process and manage log data. For instance, the Elastic stack (https://www.elastic.co/what-is/elk-stack) (a.k.a. “ELK” stack) is a popular option to collect, process, and analyze log data (possibly from different sources) in a centralized manner.

Figure 1 provides an overview about how the life-cycle of log data relates to different stages of the development cycle. First, the developer instruments the source with API calls to a logging framework (e.g., SLF4J or Log4J) to record events about the internal state of the system (in this case, whenever the reference “ data ” is “null”). Once the system is live in production, it generates data continuously whenever the execution flow reaches the log statements. The data provided by application logs (i.e., data generated from API calls of logging frameworks) and runtime environments (e.g., CPU and disk usage) are essential to detect and diagnose undesired behavior and improve software reliability. In practice, companies rely on a logging infrastructure to process and manage that data. In the context of the Elastic stack, possible components would be Elasticsearch (https://www.elastic.co/elasticsearch/), Logstash (https://www.elastic.co/logstash) and Kibana (https://www.elastic.co/kibana): Logstash is a log processor tool with several plugins available to parse and extract log data, Kibana provides an interface for visualization, query, and exploration of log data, and Elasticsearch, the core component of the Elastic stack, is a distributed and fault-tolerant search engine built on top of Apache Lucene (https://lucene.apache.org). Variants of those components from other vendors include Grafana (https://grafana.com) for user interface and Fluentd (https://www.fluentd.org) for log processing. Once the data is available, operations engineers use dashboards to analyze trends and query the data (“Log Analysis”).

Figure 1 Overview of the life-cyle of log data.

Unfortunately, despite the rich ecosystem around industry-ready log solutions, monitoring complex systems and getting insights from log data is challenging. For instance, developers need to make several decisions that affect the quality and usefulness of log data, e.g., where to place log statements and what information should be included in the log message. In addition, log data can be voluminous and heterogeneous due to how individual developers instrument an application and also the variety in a software stack that compose a system. Those characteristics of log data make it exceedingly hard to make optimal use of log data at scale. In addition, companies need to consider privacy, retention policies, and how to effectively get value from data. Even with the support of machine learning and growing adoption of big data platforms, it is challenging to process and analyze data in a costly and timely manner.

The research community, including practitioners, have been actively working to address the challenges related to the typical life-cycle of log, i.e., how to effectively provide better tooling support for logging decisions to developers (“Logging”), how to effectively process and store log data (“Logging Infrastructure”), and how to extract insights from log data (“Log Analysis”). Previously, Rong et al. (2017) conducted a survey involving 41 primary studies to understand what was the current state of logging practices. They focused their analysis on studies addressing development practices of logging. El-Masri et al. (2020) conducted an in-depth analysis of 11 log parsing (referred as “log abstraction”) techniques and proposed a quality model based on a survey of 89 primary studies on log analysis and parsing. While these are useful, no overview exists that includes other important facets of log analysis (e.g., log analysis for quality assurance), connects the different log-related areas, and identifies the most pressing open challenges. This in-breadth knowledge is key not only to provide directions to the research community but also to bridge the gap between the different research areas, and to summarize the literature for easy access to practitioners.

In this paper, we propose a systematic mapping of the logging research area. To that aim, we study 108 papers that appeared in top-level peer-reviewed conferences and journals from different communities (e.g., machine learning, software engineering, and systems). We structure the research field in light of the life-cycle of log data, elaborate the focus of each research area, and discuss opportunities and directions for future work. Our analysis shows that (1) logging is a challenge not only in open-source projects but also in industry, (2) machine learning is a promising approach to enable contextual analysis of source code for log recommendations but further investigation is required to assess the usability of those tools in practice, (3) few studies address efficient persistence of log data, and (4) while log analysis is mature field with several applications (e.g., quality assurance and failure prediction), there are open opportunities to analyze application logs and to evaluate state-of-the-art techniques in a DevOps context.

Survey methodology

The goal of this paper is to discover, categorize, and summarize the key research results in log-based software monitoring. To this end, we perform a systematic mapping study to provide a holistic view of the literature in logging and automated log analysis. Concretely, we investigate the following research questions:RQ1: What are the publication trends in research on log-based monitoring over the years?

RQ2: What are the different research scopes of log-based monitoring?

The first research question (RQ1) addresses the historical growth of the research field. Answering this research question enables us to identify the popular venues and the communities (e.g., Software Engineering, Distributed Systems) that have been focusing on log-based monitoring innovation. Furthermore, we aim at investigating the participation of industry in the research field. Researchers can benefit from our analysis by helping them to make a more informed decision regarding venues for paper submission. In addition, our analysis also serves as a guide to practitioners willing to engage with the research community either by attending conferences or looking for references to study and experiment. The second research question (RQ2) addresses the actual mapping of the primary studies. As illustrated in Fig. 1, the life-cycle of log data contains different inter-connected contexts (i.e., “Logging”, “Log Infrastructure”, and “Log Analysis”) with their own challenges and concerns that span the entire development cycle. Answering this research question enables us to identify those concerns for each context and quantify the research effort by the number of primary studies in a particular category. In addition, we aim at providing an overview of the studies so practitioners and researchers are able to use our mapping study as a starting point for an in-depth analysis of a particular topic of interest.

Overall, we follow the standard guidelines for systematic mapping (Petersen et al., 2008). Our survey methodology is divided into four parts as illustrated in Fig. 2. First, we perform preliminary searches to derive our search criteria and build an initial list of potential relevant studies based on five data sources. Next, we apply our inclusion/exclusion criteria to arrive at the eventual list of selected papers up to 2018 (when we first conducted the survey). We then conduct the data extraction and classification procedures. Finally, we update the results of our survey to include papers published in 2019.

Figure 2 Overview of survey methodology: our four steps consists of the discovery of related studies (“Search Process”), the selection of relevant studies (“Study Selection”), the mapping process (“Classification”), and the update for papers published in 2019 (“Survey Update”).

Data sources and search process

To conduct our study, we considered five popular digital libraries from different publishers based on other literature reviews in software engineering, namely, ACM Digital Library, IEEE Xplore, SpringerLink, Scopus, and Google Scholar. By considering five digital libraries, we maximize the range of venues and increase the diversity of studies related to logging. In addition, this decision reduces the bias caused by the underlying search engine since two digital libraries may rank the results in a different way for the same equivalent search.

We aim to discover relevant papers from different areas as much as possible. However, it is a challenge to build an effective query for the five selected digital libraries without dealing with a massive amount of unrelated results, since terms such as “log” and “log analysis” are pervasive in many areas. Conversely, inflating the search query with specific terms to reduce false positives would bias our study to a specific context (e.g., log analysis for debugging). To find a balance between those cases, we conducted preliminary searches with different terms and search scopes, e.g., full text, title, and abstract. We considered terms based on “log”, its synonyms, and activities related to log analysis. During this process, we observed that forcing the presence of the term “log” helps to order relevant studies on the first pages. In case the data source is unable to handle word stemming automatically (e.g., “log” and “logging”), we enhance the query with the keywords variations. In addition, configured the data sources to search on titles and abstracts whenever it was possible. In case the data source provides no support to search on titles and abstracts, we considered only titles to reduce false positives. This process resulted in the following search query:

log AND (trace OR event OR software OR system OR code OR detect OR mining OR analysis OR monitoring OR web OR technique OR develop OR pattern OR practice)

Dealing with multiple libraries requires additional work to merge data and remove duplicates. In some cases, the underlying information retrieval algorithms yielded unexpected results when querying some libraries, such as duplicates within the data source and entries that mismatch the search constraints. To overcome those barriers, we implemented auxiliary scripts to cleanup the dataset. We index the entries by title to eliminate duplicates, and we remove entries that fail to match the search criteria. Furthermore, we keep the most recent work when we identify two entries with the same title and different publication date (e.g., journal extension from previous work).

As of December of 2018, when we first conducted this search, we extracted 992 entries from Google Scholar, 1,122 entries from ACM Digital Library, 1,900 entries from IEEE Xplore, 2,588 entries from Scopus, and 7,895 entries from SpringerLink (total of 14,497 entries). After merging and cleaning the data, we ended up with 4,187 papers in our initial list.

Study selection

We conduct the selection process by assessing the 4,187 entries according to inclusion/exclusion criteria and by selecting publications from highly ranked venues. We define the criteria as follows:C1: It is an English manuscript.

C2: It is a primary study.

C3: It is a full research paper accepted through peer-review.

C4: The paper uses the term “log” in a software engineering context, i.e., logs to describe the behavior of a software system. We exclude papers that use the term “log” in an unrelated semantic (e.g., deforestation, life logging, well logging, log function).

The rationale for criterion C1 is that major venues use English as the standard idiom for submission. The rationale for criterion C2 is to avoid including secondary studies in our mapping, as suggested by Kitchenham & Charters (2007). In addition, the process of applying this criterion allows us to identify other systematic mappings and systematic literature reviews related to ours. The rationale for criterion C3 is that some databases return gray literature as well as short papers; our focus is on full peer-reviewed research papers, which we consider mature research, ready for real-world tests. Note that different venues might have different page number specifications to determine whether a submission is a full or short paper, and these specifications might change over time. We consulted the page number from each venue to avoid unfair exclusion. The rationale for criterion C4 is to exclude papers that are unrelated to the scope of this mapping study. We noticed that some of the results are in the context of, e.g., mathematics and environmental studies. While we could have tweaked our search criteria to minimize the occurrence of those false positives (e.g., NOT deforestation), we were unable to systematically derive all keywords to exclude; therefore, we favored higher false positive rate in exchange of increasing the chances of discovering relevant papers.

The first author manually performed the inclusion procedure. He analyzed the title and abstracts of all the papers marking the paper as “in” or “out”. During this process, the author applied the criteria and categorized the reasons for exclusion. For instance, whenever an entry fails the criteria C4, the authors classified it as “Out of Scope”. The categories we used are: “Out of Scope”, “Short/workshop paper”, “Not a research paper”, “Unpublished” (e.g., unpublished self-archived paper indexed by Google Scholar), “Secondary study”, and “Non-English manuscript”. It is worth mentioning that we flagged three entries as “Duplicate” as our merging step missed these cases due to special characters in the title. After applying the selection criteria, we removed 3,872 entries resulting in 315 entries.

In order to filter the remaining 315 papers by rank, we used the CORE Conference Rank (CORE Rank) (http://www.core.edu.au/conference-portal) as a reference. We considered studies published only in venues ranked as A* or A. According to the CORE Rank, those categories indicate that the venue is widely known in the computer science community and has a strict review process by experienced researches. After applying the rank criteria, we removed 219 papers.

Our selection consists of (315 − 219 =) 96 papers after applying inclusion/exclusion criteria (step 1) and filtering by venue rank (step 2). Table 1 summarises the selection process.

Table 1 Distribution of study selection when the survey was first conducted.

Selection Step	Qty	
Step 1. Exclusion by selection criteria	3,872	
Out of scope (failed C4)	3,544	
Short/workshop paper (failed C3)	276	
Not a research paper (failed C3)	40	
Non-English manuscript (failed C1)	4	
Unpublished (failed C3)	3	
Duplicate	3	
Secondary study (failed C2)	2	
Preliminary inclusion of papers	315	
Step 2. Exclusion by venue rank (neither A* nor A)	219	
Unranked	143	
Rank B	47	
Rank C	30	
Inclusion of papers (up to 2018, inclusive)	96	

Data extraction and classification

We focus the data extraction process to the required data to answer our research questions.

To answer RQ1, we collect metadata from the papers and their related venues. Concretely, we define the following schema: “Year of publication”, “Type of publication”, “Venue name”, and “Research community”. The fields “Year of publication” and “Venue name” are readly available on the scrapped data from the data sources. To extract the field “Type of publication”, we automatically assign the label “journal” if it is a journal paper. For conference papers, we manually check the proceedings to determine if it is a “research track” or “industry track” paper (we assume “research track” if not explicitly stated). To extract the field “Research community”, we check the topics of interest from the conferences and journals. This information is usually available in a “call for papers” page. Later, we manually aggregate the venues and we merge closely related topics (e.g., Artificial Intelligence, Machine Learning, and Data Science). While a complete meta-analysis is out of scope from our study, we believe the extracted data is sufficient to address the research question. Figure 3 summarizes the process for RQ1.

Figure 3 Data extraction for RQ1.

To answer RQ2, we collect the abstracts from the primary studies. In this process, we structure the abstract to better identify the motivation of the study, what problem the authors are addressing, how the researchers are mitigating the problem, and the results of the study. Given the diverse set of problems and domains, we first group the studies according to their overall context (e.g., whether the paper relates to “Logging”, “Log Infrastructure”, or “Log Analysis”). To mitigate self-bias, we conducted two independent triages and compared our results. In case of divergence, we review the paper in depth to assign the context that better fits the paper. To derive the classification schemafor each context, we perform the keywording of abstracts (Petersen et al., 2008). In this process, we extract keywords in the abstract (or introduction, if necessary) and cluster similar keywords to create categories. We perform this process using a random sample of papers to derive an initial classification schema.

Later, with all the papers initially classified, the authors explored the specific objectives of each paper and review the assigned category. To that aim, the first and second authors performed card sorting (Spencer & Warfel, 2004; Usability.gov, 2019) to determine the goal of each of the studied papers. Note that, in case new categories emerge in this process, we generalize them in either one of the existing categories or enhance our classification schema to update our view of different objectives in a particular research area. After the first round of card sorting, we noticed that some of the groups (often the ones with high number of papers) could be further broken down in subcategories (we discuss the categories and related subcategories in the Results section).

The first author conducted two separate blinded classifications on different periods of time to measure the degree of adherence to the schema given that classification is subject of interpretation, and thus, a source of bias. The same outcome converged on 83% of the cases (80 out of the 96 identified papers). The divergences were then discussed with the second author of this paper. Furthermore, the second author reviewed the resulting classification. Note that, while a paper may address more than one category, we choose the category related to the most significant contribution of that paper. Figure 4 summarizes the process for RQ2.

Figure 4 Data extraction and classification for RQ2. The dashed arrows denote the use of the data schema by the researchers with the primary studies.

Survey update

As of October of 2020, we updated our survey to include papers published in 2019 since we first conducted this analysis during December in 2018. To this end, we select all 11 papers from 2018 and perform forward snowballing to fetch a preliminary list of papers from 2019. We use snowballing for simplicity since we can leverage the “Cited By” feature from Google Scholar rather than scraping data of all five digital libraries. It is worth mentioning that we limit the results up to 2019 to avoid incomplete results for 2020.

For the preliminary list of 2019, we apply the same selection and rank criteria (see Section “Study Selection”); then, we analyze and map the studies according to the existing classification schema (see Section “Data Extraction and Classification”). In this process, we identify 12 new papers and merge them with our existing dataset. Our final dataset consists of (96 + 12 =) 108 papers.

Results

Publication trends (RQ1)

Figure 5 highlights the growth of publication from 1992 to 2019. The interest on logging has been continuously increasing since the early 2000’s. During this time span, we observed the appearance of industry track papers reporting applied research in a real context. This gives some evidence that the growing interest on the topic attracted not only researchers from different areas but also companies, fostering the collaboration between academia and industry.

Figure 5 Growth of publication types over the years.

Labels indicate the number of publication per type in a specific year. There are 108 papers in total.

We identified 108 papers (72 research track papers, 24 journals, and 12 industry track papers) published in 46 highly ranked venues spanning different communities (Table 2). Table 2 highlights the distribution of venues grouped by the research community, e.g., there are 44 papers published on 10 Software Engineering venues.

Table 2 Distribution of venues and publications grouped by research communities.

Research community	# of venues	# of papers	
Software Engineering	10	44	
Distributed Systems and Cloud Computing	10	20	
Systems	9	17	
Artificial Intelligence, Machine Learning, and Data Science (AI)	8	13	
Security	5	7	
Information Systems	3	6	
Databases	1	1	
Total	46	108	

Table 3 highlights the most recurring venues in our dataset (we omitted venues with less than three papers for brevity). The “International Conference on Software Engineering (ICSE)”, the “Empirical Software Engineering Journal (EMSE)”, and the “International Conference on Dependable Systems and Networks (DSN)” are the top three recurring venues related to the subject and are well-established venues. DSN and ICSE are conferences with more than 40 editions each and EMSE is a journal with an average of five issues per year since 1996. At a glance, we noticed that papers from DSN have an emphasis on log analysis of system logs while papers from ICSE and EMSE have an emphasis on development aspects of logging practices (more details about the research areas in the next section). Note that Table 3 also concentrates 65% (71 out of 108) of the primary studies in our dataset.

Table 3 Top recurring venues ordered by number of papers.

There are 14 (out of 46) recurring venues with at least three papers published (omitted venues with less than three papers for brevity).

Venue (acronym)	References	Qty	
		
International Conference on Software Engineering (ICSE)	Andrews & Zhang (2003), Yuan, Park & Zhou (2012), Beschastnikh et al. (2014), Fu et al. (2014a), Pecchia et al. (2015), Zhu et al. (2015), Lin et al. (2016), Chen & Jiang (2017a), Li et al. (2019b), Zhu et al. (2019)	10	
Empirical Software Engineering Journal (EMSE)	Huynh & Miller (2009), Shang, Nagappan & Hassan (2015), Russo, Succi & Pedrycz (2015), Chen & Jiang (2017b), Li, Shang & Hassan (2017), Hassani et al. (2018), Li et al. (2018), Zeng et al. (2019), Li et al. (2019a)	9	
IEEE/IFIP International Conference on Dependable Systems and Networks (DSN)	Oliner & Stearley (2007), Lim, Singh & Yajnik (2008), Cinque et al. (2010), Di Martino, Cinque & Cotroneo (2012), El-Sayed & Schroeder (2013), Oprea et al. (2015), He et al. (2016a), Neves, Machado & Pereira (2018)	8	
International Symposium on Software Reliability Engineering (ISSRE)	Tang & Iyer (1992), Mariani & Pastore (2008), Banerjee, Srikanth & Cukic (2010), Pecchia & Russo (2012), Farshchi et al. (2015), He et al. (2016b), Bertero et al. (2017)	7	
International Conference on Automated Software Engineering (ASE)	Andrews (1998), Chen et al. (2018), He et al. (2018a), Ren et al. (2019), Liu et al. (2019a)	5	
International Symposium on Reliable Distributed Systems (SRDS)	Zhou et al. (2010), Kc & Gu (2011), Fu et al. (2012), Chuah et al. (2013), Gurumdimma et al. (2016)	5	
ACM International Conference on Knowledge Discovery and Data Mining (KDD)	Makanju, Zincir-Heywood & Milios (2009), Nandi et al. (2016), Wu, Anchuri & Li (2017), Li et al. (2017)	4	
IEEE International Symposium on Cluster, Cloud and Grid Computing (CCGrid)	Prewett (2005), Yoon & Squicciarini (2014), Lin et al. (2015), Di et al. (2017)	4	
IEEE Transactions on Software Engineering (TSE)	Andrews & Zhang (2003), Tian, Rudraraju & Li (2004), Cinque, Cotroneo & Pecchia (2013), Liu et al. (2019b)	4	
Annual Computer Security Applications Conference (ACSAC)	Abad et al. (2003), Barse & Jonsson (2004), Yen et al. (2013)	3	
IBM Journal of Research and Development	Aharoni et al. (2011), Ramakrishna et al. (2017), Wang et al. (2017)	3	
International Conference on Software Maintenance and Evolution (ICSME)	Shang et al. (2014), Zhi et al. (2019), Anu et al. (2019)	3	
IEEE International Conference on Data Mining (ICDM)	Fu et al. (2009), Xu et al. (2009a), Tang & Li (2010)	3	
Journal of Systems and Software (JSS)	Mavridis & Karatza (2017), Bao et al. (2018), Farshchi et al. (2018)	3	
Total		71	

Overview of research areas (RQ2)

We grouped the studied papers among the following three categories based in our understanding about the life-cycle of log data (see Fig. 1). For each category, we derived subcatories that emerged from our keywording process (see Section “Data Extraction and Classification”):LOGGING: Research in this category aims at understanding how developers conduct logging practices and providing better tooling support to developers. There are three subcategories in this line of work: (1) empirical studies on logging practices, (2) requirements for application logs, and (3) implementation of log statements (e.g., where and how to log).

LOG INFRASTRUCTURE: Research in this category aims at improving log processing and persistence. There are two subcategories in this line of work: (1) log parsing, and (2) log storage.

LOG ANALYSIS: Research in this category aims at extracting knowledge from log data. There are eight subcategories in this line of work: (1) anomaly detection, (2) security and privacy, (3) root cause analysis, (4) failure prediction, (5) quality assurance, (6) model inference and invariant mining, (7) reliability and dependability, and (8) log platforms.

We provide an overview of the categories, their respective descriptions, and summary of our results in Table 4. In summary, we observed that LOG ANALYSIS dominates most of the research effort (68 out of 108 papers) with papers published since the early 90’s. LOG INFRASTRUCTURE is younger than LOG ANALYSIS as we observed papers starting from 2007 (16 out of 108 papers). LOGGING is the youngest area of interest with an increasing momentum for research (24 out of 108 papers). In the following, we elaborate our analysis and provide an overview of the primary studies.

Table 4 Summary of our mapping study.

The 108 papers are grouped into three main research areas, and each area has subcategories according to the focus of the study.

Category	Description	Papers	Qty	
Logging: The development of effective logging code	24	
Empirical Studies	Understanding and insights about how developers conduct logging in general	Yuan, Park & Zhou (2012), Chen & Jiang (2017b), Shang et al. (2014), Shang, Nagappan & Hassan (2015), Pecchia et al. (2015), Kabinna et al. (2016), Li et al. (2019b), Zeng et al. (2019)	8	
Log requirements	Assessment of log conformance given a known requirement	Cinque et al. (2010), Pecchia & Russo (2012), Cinque, Cotroneo & Pecchia (2013), Yuan et al. (2012), da Cruz et al. (2004)	5	
Implementation of log statements	Focus on what to log, where to log, and how to log	Chen & Jiang (2017a), Hassani et al. (2018), Fu et al. (2014a), Zhu et al. (2015), Li et al. (2018), Li, Shang & Hassan (2017), He et al. (2018a), Li et al. (2019a), Liu et al. (2019b), Anu et al. (2019), Zhi et al. (2019)	11	
Log Infrastructure: Techniques to enable and fulfil the requirements of the analysis process	16	
Parsing	Extraction of log templates from raw log data	Aharon et al. (2009), Makanju, Zincir-Heywood & Milios (2009), Makanju, Zincir-Heywood & Milios (2012), Liang et al. (2007), Gainaru et al. (2011), Hamooni et al. (2016), Zhou et al. (2010), Lin et al. (2016), Tang & Li (2010), He et al. (2016a), He et al. (2018b), Zhu et al. (2019), Agrawal, Karlupia & Gupta (2019)	13	
Storage	Efficient persistence of large datasets of logs	Lin et al. (2015), Mavridis & Karatza (2017), Liu et al. (2019a)	3	
Log Analysis: Insights from processed log data	68	
Anomaly detection	Detection of abnormal behaviour	Tang & Iyer (1992), Oliner & Stearley (2007), Lim, Singh & Yajnik (2008), Xu et al. (2009b), Xu et al. (2009a), Fu et al. (2009), Ghanbari, Hashemi & Amza (2014), Gao et al. (2014), Juvonen, Sipola & Hämäläinen (2015), Farshchi et al. (2015), He et al. (2016b), Nandi et al. (2016), Du et al. (2017), Bertero et al. (2017), Lu et al. (2017), Debnath et al. (2018), Bao et al. (2018), Farshchi et al. (2018), Zhang et al. (2019), Meng et al. (2019)	20	
Security and privacy	Intrusion and attack detection	Oprea et al. (2015), Chu et al. (2012), Yoon & Squicciarini (2014), Yen et al. (2013), Barse & Jonsson (2004), Abad et al. (2003), Prewett (2005), Butin & Le Métayer (2014), Goncalves, Bota & Correia (2015)	9	
Root cause analysis	Accurate failure identification and impact analysis	Gurumdimma et al. (2016), Kimura et al. (2014), Pi et al. (2018), Chuah et al. (2013), Zheng et al. (2011), Ren et al. (2019)	6	
Failure prediction	Anticipating failures that leads a system to an unrecoverable state	Wang et al. (2017), Fu et al. (2014b), Russo, Succi & Pedrycz (2015), Khatuya et al. (2018), Shalan & Zulkernine (2013), Fu et al. (2012)	6	
Quality assurance	Logs as support for quality assurance activities	Andrews (1998), Andrews & Zhang (2000), Andrews & Zhang (2003), Chen et al. (2018)	4	
Model inference and invariant mining	Model and invariant checking	Ulrich et al. (2003), Mariani & Pastore (2008), Tan et al. (2010), Beschastnikh et al. (2014), Wu, Anchuri & Li (2017), Awad & Menasce (2016), Kc & Gu (2011), Lou et al. (2010), Steinle et al. (2006), Di Martino, Cinque & Cotroneo (2012)	10	
Reliability and dependability	Understand dependability properties of systems (e.g., reliability, performance)	Banerjee, Srikanth & Cukic (2010), Tian, Rudraraju & Li (2004), Huynh & Miller (2009), El-Sayed & Schroeder (2013), Ramakrishna et al. (2017), Park et al. (2017)	6	
Log platforms	Full-fledged log analysis platforms	Li et al. (2017), Aharoni et al. (2011), Yu et al. (2016), Balliu et al. (2015), Di et al. (2017), Neves, Machado & Pereira (2018), Gunter et al. (2007)	7	

Logging

Log messages are usually in the form of free text and may expose parts of the system state (e.g., exceptions and variable values) to provide additional context. The full log statement also includes a severity level to indicate the purpose of that statement. Logging frameworks provide developers with different log levels: debug for low level logging, info to provide information on the system execution, error to indicate unexpected state that may compromise the normal execution of the application, and fatal to indicate a severe state that might terminate the execution of the application. Logging an application involves several decisions such as what to log. These are all important decisions since they have a direct impact on the effectiveness of the future analysis. Excessive logging may cause performance degradation due the number of writing operations and might be costly in terms of storage. Conversely, insufficient information undermines the usefulness of the data to the operations team. It is worth mentioning that the underlying environment also provides valuable data. Environment logs provide insights about resource usage (e.g., CPU, memory and network) and this data can be correlated with application logs on the analysis process. In contrast to application logs, developers are often not in control of environment logs. On the other hand, they are often highly structured and are useful as a complementary data source that provides additional context.

LOGGING deals with the decisions from the developer’s perspective. Developers have to decide the placement of log statements, what message description to use, which runtime information is relevant to log (e.g., the thrown exception), and the appropriate severity level. Efficient and accurate log analysis rely on the quality of the log data, but it is not always feasible to know upfront the requirements of log data during development time.

We observed three different subcategories in log engineering: (1) empirical studies on log engineering practices, (2) techniques to improve log statements based on known requirements for log data, and (3) techniques to help developers make informed decisions when implementing log statements (e.g., where and how to log). In the following, we discuss the 24 log engineering papers in the light of these three types of studies.

Empirical studies

Understanding how practitioners deal with the log engineering process in a real scenario is key to identify open problems and provide research directions. Papers in this category aim at addressing this agenda through empirical studies in open-source projects (and their communities).

Yuan, Park & Zhou (2012) conducted the first empirical study focused on understanding logging practices. They investigated the pervasiveness of logging, the benefits of logging, and how log-related code changes over time in four open-source projects (Apache httpd, OpenSSH, PostgresSQL, and Squid). In summary, while logging was widely adopted in the projects and were beneficial for failure diagnosis, they show that logging as a practice relies on the developer’s experience. Most of the recurring changes were updates to the content of the log statement.

Later, Chen & Jiang (2017b) conducted a replication study with a broader corpus: 21 Java-based projects from the Apache Foundation. Both studies confirm that logging code is actively maintained and that log changes are recurrent; however, the presence of log data in bug reports are not necessarily correlated to the resolution time of bug fixes (Chen & Jiang, 2017b). This is understandable as resolution time also relates to the complexity of the reported issue.

It is worth mentioning that the need for tooling support for logging also applies in an industry setting. For instance, in a study conducted by Pecchia et al. (2015), they show that the lack of format conventions in log messages, while not severe for manual analysis, undermines the use of automatic analysis. They suggest that a tool to detect inconsistent conventions would be helpful for promptly fixes. In a different study, Zhi et al. (2019) analyses log configurations on 10 open-source projects and 10 Alibaba systems. They show that developers often rely on logging configurations to control the throughput of data and quality of data (e.g., suppressing inconvenient logs generated from external dependencies, changing the layout format of the recorded events) but finding optimal settings is challenging (observed as recurrent changes on development history).

In the context of mobile development, Zeng et al. (2019) show that logging practices are different but developers still struggle with inconsistent logging. They observed a lower density of log statements compared to previous studies focused on server and desktop systems (Chen & Jiang, 2017b; Yuan, Park & Zhou, 2012) by analyzing +1.4K Android apps hosted on F-Droid. Logging practices in mobile development differ mainly because developers need to consider the overhead impact on user’s device. The authors observed a statistically significant difference in terms of response time, battery consumption, and CPU when evaluating eight apps with logging enabled and disabled.

Understanding the meaning of logs is important not only for analysis but also for maintenance of logging code. However, one challenge that developers face is to actively update log-related code along functionalities. The code base naturally evolves but due to unawareness on how features are related to log statements, the latter become outdated and may produce misleading information (Yuan, Park & Zhou, 2012; Chen & Jiang, 2017b). This is particularly problematic when the system is in production and developers need to react for user inquiries. In this context, Shang et al. (2014) manually analyzed mailing lists and sampled log statements from three open-source projects (Apache Hadoop, Zookeper, and Cassandra) to understand how practitioners and customers perceive log data. They highlight that common inquiries about log data relate to the meaning, the cause, the context (e.g., in which cases a particular message appears in the log files), the implications of a message, and solutions to manifested problems.

In a different study, Shang, Nagappan & Hassan (2015) investigated the relationship between logging code and the overall quality of the system though a case study on four releases from Apache Hadoop and JBoss. They show that the presence of log statements are correlated to unstable source files and are strong indicators of defect-prone features. In other words, classes that are more prone to defects often contain more logs.

Finally, Kabinna et al. (2016) explored the reasons and the challenges of migrating to a different logging library. The authors noticed that developers have different drivers for such a refactoring, e.g., to increase flexibility, performance, and to reduce maintenance effort. Interestingly, the authors also observed that most projects suffer from post-migration bugs because of the new logging library, and that migration rarely improved performance.

Log requirements

An important requirement of log data is that it must be informative and useful to a particular purpose. Papers in this subcategory aim at evaluating whether log statements can deliver expected data, given a known requirement.

Fault injection is a technique that can be useful to assess the diagnosibility of log data, i.e., whether log data can manifest the presence of failures. Past studies conducted experiments in open-source projects and show that logs are unable to produce any trace of failures in most cases (Cinque et al., 2010; Pecchia & Russo, 2012; Cinque, Cotroneo & Pecchia, 2013). The idea is to introduce faults in the system under test, run tests (these have to manifest failures), and compare the log data before and after the experiment. Examples of introduced faults are missing method calls and missing variable assignment. The authors suggest the usage of fault injection as a guideline to identify and add missing log statements.

Another approach to address the diagnosability in log data was proposed by Yuan et al. (2012). LOGENHANCER leverages program analysis techniques to capture additional context to enhance log statements in the execution flow. Differently from past work with fault injection, LOGENHANCER proposes the enhancement of existing log statements rather than addition of log statements in missing locations.

In the context of web services, da Cruz et al. (2004) already explored the idea of enhancing log data. An interesting remark pointed by the authors is that, in the context of complex system with third-party libraries, there is no ownership about the format and content of log statements. This is an issue if the log data generated is inappropriate and requires changes (as observed by Zhi et al. (2019)). To overcome this issue, they propose WSLOG A, a logging framework based on SOAP intermediaries that intercepts messages exchanged between client and server and enhances web logs with important data for monitoring and auditing, e.g., response and processing time.

Implementation of log statements

Developers need to make several decisions at development time that influence the quality of the generated log data. Past studies in logging practices show that in practice, developers rely on their own experience and logging is conducted in a trial-and-error manner in open-source projects (Yuan, Park & Zhou, 2012; Chen & Jiang, 2017b) and industry (Pecchia et al., 2015). Papers in this subcategory aim at studying logging decisions, i.e., where to place log statements, which log level to use, and how to write log messages.

Hassani et al. (2018) proposed a set of checkers based in an empirical study of log-related changes in two open-source projects (Apache Hadoop and Apache Camel). They observed that typos in log messages, missing guards (i.e., conditional execution of log statement according to the appropriate level), and missing exception-related logging (e.g., unlogged exception or missing the exception in a log statement) are common causes for code changes. Li et al. (2019a) also analyze log changes across several revisions on 12 C/C++ open-source projects. However, they mine rules based on the type of modification (e.g., update on log descriptor) and contextual characteristics from the revision. The rational is that new code changes with similar contextual characteristics should have similar type of log modification. The authors proposed this method in the form of a tool named LOGTRACKER. In another study, Chen & Jiang (2017a) analyzed 352 pairs of log-related changes from ActiveMQ, Hadoop, and Maven (all Apache projects), and proposed LCANALYZER, a checker that encodes the anti-patterns identified on their analysis. Some of these patterns are usage of nullable references, explicit type cast, and malformed output (e.g., referencing data types without user-friendly string representation) in the log statement. Li et al. (2019b) addressed additional anti-patterns caused mainly by improper copy-and-paste, e.g., same log statement reused on different catch blocks. They derived five duplication anti-patterns by studying 3K duplicated log statements on Hadoop, ElasticSearch, CloudStack, and Cassandra, and encoded those anti-patterns in a checker named DLFINDER. On the evaluation, they discovered not only new issues on the analyzed systems but also on other two systems (Camel and Wicket). Note that several recurrent problems aforementioned can be capture by static analysis before merging changes into the code base.

Deciding where to place log statements is critical to provide enough context for later analysis. One way to identify missing locations is to use fault injection (see “Log Requirements”). However, the effectiveness of that approach is limited to the quality of tests and the ability of manifesting failures. Furthermore, log placement requires further contextual information that is unfeasible to capture only with static analysis. Another approach to address consistent log placement in large code bases is to leverage source code analysis and statistical models to mine log patterns. Fu et al. (2014a) conducted an empirical study in two Microsoft C# systems and proposed five classifications for log placement: three for unexpected situations and two for regular monitoring. Unexpected situations cover log statements triggered by failed assertions (“assertion logging”), exception handling or throw statements (“exception logging”), and return of unexpected values after a checking condition (“return-value-check logging”). Regular monitoring cover the remaining cases of log statements that can be in logic branches (“logic-branch logging”) or not (“observing-point logging”). Later, Zhu et al. (2015) proposed LOGADVISOR, a technique that leverages supervised learning with feature engineering to suggest log placement for unexpected situations, namely catch blocks (“exception logging”) and if blocks with return statements (“return-value-check logging”). Some of the features defined for the machine learning process are size of the method, i.e., number of lines of source code, name of method parameters, name of local variables, and method signature. They evaluated LOGADVISOR on two proprietary systems from Microsoft and two open-source projects hosted on GitHub. The results indicate the feasibility of applying machine learning to provide recommendations for where to place new log statements. Li et al. (2018) approached the placement problem by correlating the presence of logging code with the context of the source code. The rationale is that some contexts (defined through topic models) are more likely to contain log statements (e.g., network or database operations) than others (e.g., getter methods). In this work, the authors analyze log placement at method level rather than block-level as in previous work (Fu et al., 2014a; Zhu et al., 2015).

Choosing the appropriate severity level of log statements is a challenge. Recall that logging frameworks provide the feature of suppressing log messages according to the log severity. Li, Shang & Hassan (2017) proposed a machine learning-based technique to suggest the log level of a new log statement. The underlying model uses ordinal regression, which is useful to predict classes, i.e., log level, but taking into account their severity order, e.g., info < warning < error. Their technique provides better accuracy than random guessing and guessing based on the distribution of log levels in the source code. They report that the log message and the surrounding context of the log statement are good predictors of the log level. It is worth mentioned that Hassani et al. (2018) also addressed the problem of identifying appropriate log level in their study on log-related changes by examining the entropy of log messages and log levels. The underlying idea is that log levels that are commonly associated with a log message also should be used on other log statements with similar log messages. While this approach is intuitive and precise, the authors report low recall. Both studies highlight the relationship of the log message and associate severity of a log statement. In another study, Anu et al. (2019) also proposes a classifier for log level recommendation. They focus on log statements located on if-else blocks and exception handling. In terms of feature engineering, the authors leverage mostly the terms associated in the code snippet (e.g., log message, code comments, and method calls) while Li, Shang & Hassan (2017) use quantitative metrics extracted from code (e.g., length of log message and code complexity). However it remains open how both techniques compare in terms of performance.

An important part of log statements is the description of the event being logged. Inappropriate descriptions are problematic and delay the analysis process. He et al. (2018a) conducted an empirical study focused on what developers log. They analyzed 17 projects (10 in Java and 7 in C#) and concluded that log descriptors are repetitive and small in vocabulary. For this reason, they suggest that it is feasible to exploit information retrieval methods to automatically generate log descriptions.

In addition to log descriptors, the state of the system is another important information the event being logged. Liu et al. (2019b) proposed a machine learning-based approach to aid developers about which variables to log based on the patterns of existing log statements. The technique consists of four layers: embedding, Recurrent Neural Network (RNN), self-attention mechanism, and output. Results indicate better performance than random guess and information retrieve approaches on the evaluation of nine Java projects.

Log infrastructure

The infrastructure supporting the analysis process plays an important role because the analysis may involve the aggregation and selection of high volumes of data. The requirements for the data processing infrastructure depend on the nature of the analysis and the nature of the log data. For instance, popular log processors, e.g., Logstash and Fluentd, provide regular expressions out-of-the-box to extract data from well-known log formats of popular web servers (e.g., Apache Tomcat and Nginx). However, extracting content from highly unstructured data into a meaningful schema is not trivial.

LOG INFRASTRUCTURE deals with the tooling support necessary to make the further analysis feasible. For instance, data representation might influence on the efficiency of data aggregation. Other important concerns include the ability of handling log data for real-time or offline analysis and scalability to handle the increasing volume of data.

We observed two subcategories in this area: (1) log parsing, and (2) log storage. In the following, we summarize the 16 studies on log infrastructure grouped by these two categories.

Log parsing

Parsing is the backbone of many log analysis techniques. Some analysis operate under the assumption that source-code is unavailable; therefore, they rely on parsing techniques to process log data. Given that log messages often have variable content, the main challenge tackled by these papers is to identify which log messages describe the same event. For example, “Connection from A port B” and “Connection from C port D” represent the same event. The heart of studies in parsing is the template extraction from raw log data. Fundamentally, this process consists of identifying the constant and variable parts of raw log messages.

Several approaches rely on the “textual similarity” between the log messages. Aharon et al. (2009) create a dictionary of all words that appear in the log message and use the frequency of each word to cluster log messages together. Somewhat similar, IPLOM (Iterative Partitioning Log Mining) leverages the similarities between log messages related to the same event, e.g., number, position, and variability of tokens (Makanju, Zincir-Heywood & Milios, 2009; Makanju, Zincir-Heywood & Milios, 2012). Liang et al. (2007) also build a dictionary out of the keywords that appear in the logs. Next, each log is converted to a binary vector, with each element representing whether the log contains that keyword. With these vectors, the authors compute the correlation between any two events.

Somewhat different from others, Gainaru et al. (2011) cluster log messages by searching for the best place to split a log message into its “constant” and its “variable” parts. These clusters are self-adaptive as new log messages are processed in a streamed fashion. Hamooni et al. (2016) also uses string similarity to cluster logs. Interestingly, authors however made use of map-reduce to speed up the processing. Finally, Zhou et al. (2010) propose a fuzzy match algorithm based on the contextual overlap between log lines.

Transforming logs into “sequences” is another way of clustering logs. Lin et al. (2016) convert logs into vectors, where each vector contains a sequence of log events of a given task, and each event has a different weight, calculated in different ways. Tang & Li (2010) propose LOGTREE, a semi-structural way of representing a log message. The overall idea is to represent a log message as a tree, where each node is a token, extracted via a context-free grammar parser that the authors wrote for each of the studied systems. Interestingly, in this paper, the authors raise awareness to the drawbacks of clustering techniques that consider only word/term information for template extraction. According them, log messages related to same events often do not share a single word.

From an empirical perspective, He et al. (2016a) compared four log parsers on five datasets with over 10 million raw log messages and evaluated their effectiveness in a real log-mining task. The authors show, among many other findings, that current log parsing methods already achieve high accuracy, but do not scale well to large log data. Later, Zhu et al. (2019) extended the former study and evaluated a total of 13 parsing techniques on 16 datasets. In a different study, He et al. (2018b) also compared existing parsing techniques and proposed a distributed parsing technique for large-scale datasets on top of Apache Spark. The authors show that for small datasets, the technique underperforms due to the communication overhead between workers; however, for large-scale datasets (e.g., 200 million log messages), the approach overcomes traditional techniques. It is worth mentioning that the large-scale datasets were synthetically generated on top of two popular datasets due to the lack of real-world datasets. Agrawal, Karlupia & Gupta (2019) also proposes a distributed approach based on Apache Spark for distributed parsing. The comparison between the two approaches (He et al., 2018b; Agrawal, Karlupia & Gupta, 2019) remains open.

Log storage

Modern complex systems easily generate giga- or petabytes of log data a day. Thus, in the log data life-cycle, storage plays an important role as, when not handled carefully, it might become the bottleneck of the analysis process. Researchers and practitioners have been addressing this problem by offloading computation and storage to server farms and leveraging distributed processing.

Mavridis & Karatza (2017) frame the problem of log analysis at scale as a “big data” problem. Authors evaluated the performance and resource usage of two popular big data solutions (Apache Hadoop and Apache Spark) with web access logs. Their benchmarks show that both approaches scale with the number of nodes in a cluster. However, Spark is more efficient for data processing since it minimizes reads and writes in disk. Results suggest that Hadoop is better suited for offline analysis (i.e., batch processing) while Spark is better suited for online analysis (i.e., stream processing). Indeed, as mentioned early, He et al. (2018b) leverages Spark for parallel parsing because of its fast in-memory processing.

Another approach to reduce storage costs consists of data compression techniques for efficient analysis (Lin et al., 2015; Liu et al., 2019a). Lin et al. (2015) argue that while traditional data compression algorithms are useful to reduce storage footprint, the compression-decompression loop to query data undermines the efficiency of log analysis. The rationale is that traditional compression mechanisms (e.g., gzip) perform compression and decompression in blocks of data. In the context of log analysis, this results in waste of CPU cycles to compress and decompress unnecessary log data. They propose a compression approach named Cowik that operates in the granularity of log entries. They evaluated their approach in a log search and log joining system. Results suggest that the approach is able to achieve better performance on query operations and produce the same join results with less memory. Liu et al. (2019a) proposes a different approach named LOGZIP based on an intermediate representation of raw data that exploits the structure of log messages. The underlying idea is to remove redundant information from log events and compress the intermediate representation rather than raw logs. Results indicate higher compression rates compared to baseline approaches (including COWIK).

Log analysis

After the processing of log data, the extracted information serves as input to sophisticated log analysis methods and techniques. Such analysis, which make use of varying algorithms, help developers in detecting unexpected behavior, performance bottlenecks, or even security problems.

LOG ANALYSIS deals with knowledge acquisition from log data for a specific purpose, e.g., detecting undesired behavior or investigating the cause of a past outage. Extracting insights from log data is challenging due to the complexity of the systems generating that data.

We observed eight subcategories in this area: (1) anomaly detection, (2) security and privacy, (3) root cause analysis, (4) failure prediction, (5) quality assurance, (6) model inference and invariant mining, (7) reliability and dependability, and (8) platforms. In the following, we summarize the 68 studies on log analysis grouped by these seven different goals.

Anomaly detection

Anomaly detection techniques aim to find undesired patterns in log data given that manual analysis is time-consuming, error-prone, and unfeasible in many cases. We observe that a significant part of the research in the logging area is focused on this type of analysis. Often, these techniques focus on identifying problems in software systems. Based on the assumption that an “anomaly” is something worth investigating, these techniques look for anomalous traces in the log files.

Oliner & Stearley (2007) raise awareness on the need of datasets from real systems to conduct studies and provide directions to the research community. They analyzed log data from five super computers and conclude that logs do not contain sufficient information for automatic detection of failures nor root cause diagnosis, small events might dramatically impact the number of logs generated, different failures have different predictive signatures, and messages that are corrupted or have inconsistent formats are not uncommon. Many of the challenges raised by the authors are well known nowadays and have been in continuous investigation in academia.

Researchers have been trying several different techniques, such as deep learning and NLP (Du et al., 2017; Bertero et al., 2017; Meng et al., 2019; Zhang et al., 2019), data mining, statistical learning methods, and machine learning (Lu et al., 2017; He et al., 2016b; Ghanbari, Hashemi & Amza, 2014; Tang & Iyer, 1992; Lim, Singh & Yajnik, 2008; Xu et al., 2009b, Xu et al., 2009a) control flow graph mining from execution logs (Nandi et al., 2016), finite state machines (Fu et al., 2009; Debnath et al., 2018), frequent itemset mining (Lim, Singh & Yajnik, 2008), dimensionality reduction techniques (Juvonen, Sipola & Hämäläinen, 2015), grammar compression of log sequences (Gao et al., 2014), and probabilistic suffix trees (Bao et al., 2018).

Interestingly, while these papers often make use of systems logs (e.g., logs generated by Hadoop, a common case study among log analysis in general) for evaluation, we conjecture that these approaches are sufficiently general, and could be explored in (or are worth trying at on) other types of logs (e.g., application logs).

Researchers have also explored log analysis techniques within specific contexts. For instance, finding anomalies in HTTP logs by using dimensionality reduction techniques (Juvonen, Sipola & Hämäläinen, 2015), finding anomalies in cloud operations (Farshchi et al., 2015; Farshchi et al., 2018) and Spark programs (Lu et al., 2017) by using machine learning. As within many other fields of software engineering, we see an increasingly adoption of machine and deep learning. In 2016, He et al. (2016b) then evaluated six different algorithms (three supervised, and three unsupervised machine learning methods) for anomaly detection. The authors found that supervised anomaly detection methods present higher accuracy when compared to unsupervised methods; that the use of sliding windows (instead of a fixed window) can increase the accuracy of the methods; and that methods scale linearly with the log size. In 2017, Du et al. (2017) proposed DEEPLOG, a deep neural network model that used Long Short-Term Memory (LSTM) to model system logs as a natural language sequence, and Bertero et al. (2017) explored the use of NLP, considering logs fully as regular text. In 2018, Debnath et al. (2018) (by means of the LOGMINE technique (Hamooni et al., 2016)) explored the use of clustering and pattern matching techniques. In 2019, Meng et al. (2019) proposed a technique based on unsupervised learning for unstructured data. It features a transformer TEMPLATE2VEC (as an alternative to WORD2VEC) to represent extracted templates from logs and LSTMs to learn common sequences of log sequences. In addition, Zhang et al. (2019) leverages LSTM models with attention mechanism to handle unstable log data. They argue that log data changes over time due to evolution of software and models addressing log analysis need to take this into consideration.

Security and privacy

Logs can be leveraged for security purposes, such as intrusion and attacks detection.

Oprea et al. (2015) use (web) traffic logs to detect early-stage malware and advanced persistence threat infections in enterprise network, by modeling the information based on belief propagation inspired by graph theory. Chu et al. (2012) analyses access logs (in their case, from TACACS+, an authentication protocol developed by Cisco) to distinguish normal operational activities from rogue/anomalous ones. Yoon & Squicciarini (2014) focus on the analysis and detection of attacks launched by malicious or misconfigured nodes, which may tamper with the ordinary functions of the MapReduce framework. Yen et al. (2013) propose Beehive, a large-scale log analysis for detecting suspicious activity in enterprise networks, based on logs generated by various network devices. In the telecommunication context, Goncalves, Bota & Correia (2015) used clustering algorithms to identify malicious activities based on log data from firewall, authentication and DHCP servers.

An interesting characteristic among them all is that the most used log data is, by far, network data. We conjecture this is due to the fact that (1) network logs (e.g., HTTP, web, router logs) are independent from the underlying application, and that (2) network tends to be, nowadays, a common way of attacking an application.

Differently from analysis techniques where the goal is to find a bug, and which are represented in the logs as anomalies, understanding which characteristics of log messages can reveal security issues is still an open topic. Barse & Jonsson (2004) extract attack manifestations to determine log data requirements for intrusion detection. The authors then present a framework for determining empirically which log data can reveal a specific attack. Similarly, Abad et al. (2003) argue for the need of correlation data from different logs to improve the accuracy of intrusion detection systems. The authors show in their paper how different attacks are reflected in different logs, and how some attacks are not evident when analyzing single logs. Prewett (2005) examines how the unique characteristics of cluster machines, including how they are generally operated in the larger context of a computing center, can be leveraged to provide better security.

Finally, regarding privacy, Butin & Le Métayer (2014) propose a framework for accountability based on “privacy-friendly” event logs. These logs are then used to show compliance with respect to data protection policies.

Root cause analysis

Detecting anomalous behavior, either by automatic or monitoring solutions, is just part of the process. Maintainers need to investigate what caused that unexpected behavior. Several studies attempt to take the next step and provide users with, e.g., root cause analysis, accurate failure identification, and impact analysis.

Kimura et al. (2014) identify spatial-temporal patterns in network events. The authors affirm that such spatial-temporal patterns can provide useful insights on the impact and root cause of hidden network events. Ren et al. (2019) explores a similar idea in the context of diagnosing non-reproducible builds. They propose a differential analysis among different build traces based on I/O and parent-child dependencies. The technique leverages the common dependencies patterns to filter abnormal patterns and to pinpoint the cause of the non-reproducible build. Pi et al. (2018) propose a feedback control tool for distributed applications in virtualized environments. By correlating log messages and resource consumption, their approach builds relationships between changes in resource consumption and application events. Somewhat related, Chuah et al. (2013) identifies anomalies in resource usage, and link such anomalies to software failures. Zheng et al. (2011) also argue for the need of correlating different log sources for a better problem identification. In their study, authors correlate supercomputer BlueGene’s reliability, availability and serviceability logs with its job logs, and show that such a correlation was able to identify several important observations about why their systems and jobs fail. Gurumdimma et al. (2016) also leverages multiple sources of data for accurate diagnosis of malfunctioning nodes in the Ranger Supercomputer. The authors argue that, while console logs are useful for administration tasks, they can complex to analyze by operators. They propose a technique based on the correlation of console logs and resource usage information to link jobs with anomalous behavior and erroneous nodes.

Failure prediction

Being able to anticipate failures in critical systems not only represents competitive business advantage but also represents prevention of unrecoverable consequences to the business. Failure prediction is feasible once there is knowledge about abnormal patterns and their related causes. However, it differs from anomaly detection in the sense that identifying the preceding patterns of an unrecoverable state requires insights from root cause analysis. This approach shifts monitoring to a proactive manner rather than reactive, i.e., once the problem occurred.

Work in this area, as expected, relies on statistical and probabilistic models, from standard regression analysis to machine learning. Wang et al. (2017) apply random forests in event logs to predict maintenance of equipment (in their case study, ATMs). Fu et al. (2014b) use system logs (from clusters) to generate causal dependency graphs and predict failures. Russo, Succi & Pedrycz (2015) mine system logs (more specifically, sequences of logs) to predict the system’s reliability by means of linear radial basis functions, and multi-layer perceptron learners. Khatuya et al. (2018) propose ADELE, a machine learning-based technique to predict functional and performance issues. Shalan & Zulkernine (2013) utilize system logs to predict failure occurrences by means of regression analysis and support vector machines. Fu et al. (2012) also utilize system logs to predict failures by mining recurring event sequences that are correlated.

We noticed that, given that only supervised models have been used so far, feature engineering plays an important role in these papers. Khatuya et al. (2018), for example, uses event count, event ratio, mean inter-arrival time, mean inter-arrival distance, severity spread, and time-interval spread. Russo, Succi & Pedrycz (2015) use defective and non defective sequences of events as features. Shalan & Zulkernine (2013)’s paper, although not completely explicit about which features they used, mention CPU, memory utilization, read/write instructions, error counter, error messages, error types, and error state parameters as examples of features.

Quality assurance

Log analysis might support developers during the software development life cycle and, more specifically, during activities related to quality assurance.

Andrews & Zhang (2000, 2003) advocated the use of logs for testing purposes since the early 2000’s. In their work, the authors propose an approach called log file analysis (LFA). LFA requires the software under test to write a record of events to a log file, following a pre-defined logging policy that states precisely what the software should log. A log file analyzer, also written by the developers, then analyses the produced log file and only accepts it in case the run did not reveal any failures. The authors propose a log file analysis language to specify such analyses.

More than 10 years later, Chen et al. (2018) propose an automated approach to estimate code coverage via execution logs named LogCoCo. The motivation for this use of log data comes from the need to estimate code coverage from production code. The authors argue that, in a large-scale production system, code coverage from test workloads might not reflect coverage under production workload. Their approach relies on program analysis techniques to match log data and their corresponding code paths. Based on this data, LogCoCo estimates different coverage criteria, i.e., method, statement, and branch coverage. Their experiments in six different systems show that their approach is highly accurate (>96%).

Model inference and invariant mining

Model-based approaches to software engineering seek to support understanding and analysis by means of abstraction. However, building such models is a challenging and expensive task. Logs serve as a source for developers to build representative models and invariants of their systems. These models and invariants may help developers in different tasks, such as comprehensibility and testing. These approaches generate different types of models, such as (finite) state machines (Ulrich et al., 2003; Mariani & Pastore, 2008; Tan et al., 2010; Beschastnikh et al., 2014) directed workflow graphs (Wu, Anchuri & Li, 2017) client-server interaction diagrams (Awad & Menasce, 2016), invariants (Kc & Gu, 2011; Lou et al., 2010), and dependency models (Steinle et al., 2006).

State machines are the most common type of model extracted from logs. Beschastnikh et al. (2014), for example, infer state machine models of concurrent systems from logs. The authors show that their models are sufficiently accurate to help developers in finding bugs. Ulrich et al. (2003) show how log traces can be used to build formal execution models. The authors use SDL, a model-checking description technique, common in telecommunication industries. Mariani & Pastore (2008) propose an approach where state machine-based models of valid behaviors are compared with log traces of failing executions. The models are inferred via the kBehavior engine (Mariani & Pastore, 2008). Tan et al. (2010) extract state-machine views of the MapReduce flow behavior using the native logs that Hadoop MapReduce systems produce.

The mining of properties that a system should hold has also been possible via log analysis. Lou et al. (2010) derive program invariants from logs. The authors show that the invariants that emerge from their approach are able to detect numerous real-world problems. Kc & Gu (2011) aim to facilitate the troubleshooting of cloud computing infrastructures. Besides implementing anomaly detection techniques, their tool also performs invariant checks in log events, e.g., two processes performing the same task at the same time (these invariants are not automatically devised, but should be written by system administrators).

We also observe directed workflow graphs and dependency maps as other types of models built from logs. Wu, Anchuri & Li (2017) propose a method that mines structural events and transforms them into a directed workflow graph, where nodes represent log patterns, and edges represent the relations among patterns. Awad & Menasce (2016) derive performance models of operational systems based on system logs and configuration logs. Finally, Steinle et al. (2006) map dependencies among internal components through system logs, via data mining algorithms and natural language processing techniques.

Finally, and somewhat different from the other papers in this ramification, Di Martino, Cinque & Cotroneo (2012) argue that an important issue in log analysis is that, when a failure happens, multiple independent error events appear in the log. Reconstructing the failure process by grouping together events related to the same failure (also known as data coalescence techniques) can therefore help developers in finding the problem. According to the authors, while several coalescence techniques have been proposed over time (Tsao & Siewiorek, 1983; Hansen & Siewiorek, 1992), evaluating these approaches is a challenging task as the ground truth of the failure is often not available. To help researchers in evaluating their approaches, the authors propose a technique which basically generates synthetic logs along with the ground truth they represent.

Reliability and dependability

Logs can serve as a means to estimate how reliable and dependable a software system is. Research in this subcategory often focuses on large software systems, such as web and mobile applications that are distributed in general, and high performance computers.

Banerjee, Srikanth & Cukic (2010) estimate the reliability of a web Software-as-a-Service (SaaS) by analyzing its web traffic logs. Authors categorize different types of log events with different severity levels, counting, e.g, successfully loaded (non-critical) images separately from core transactions, providing different perspectives on reliability. Tian, Rudraraju & Li (2004) evaluate the reliability of two web applications, using several metrics that can be extracted from web access and error logs (e.g., errors per page hits, errors per sessions, and errors per users). The authors conclude that the usage of workload and usage patterns, present in log files, during testing phases could significantly improve the reliability of the system. Later, Huynh & Miller (2009) expanded previous work (Tian, Rudraraju & Li, 2004) by enumerating improvements for reliability assessment. They emphasize that some (http) error codes require a more in-depth analysis, e.g., errors caused by factors that cannot be controlled by the website administrators should be separated from the ones that can be controlled, and that using IP addresses as a way to measure user count can be misleading, as often many users share the same IP address.

Outside the web domain, El-Sayed & Schroeder (2013) explore a decade of field data from the Los Alamos National Lab and study the impact of different factors, such as power quality, temperature, fan activity, system usage, and even external factors, such as cosmic radiation, and their correlation with the reliability of High Performance Computing (HPC) systems. Among the lessons learned, the authors observe that the day following a failure, a node is 5 to 20 times more likely to experience an additional failure, and that power outages not only increase follow-up software failures, but also infrastructure failures, such as problems in distributed storage and file systems. In a later study, Park et al. (2017) discuss the challenges of analyzing HPC logs. Log analysis of HPC data requires understanding underlying hardware characteristics and demands processing resources to analyze and correlate data. The authors introduce an analytic framework based on NOSQL databases and Big Data technology (Spark) for efficient in-memory processing to assist system administrators.

Analyzing the performance of mobile applications can be challenging specially when they depend on back-end distributed services. IBM researchers (Ramakrishna et al., 2017) proposed MIAS (Mobile Infrastructure Analytics System) to analyze performance of mobile applications. The technique considers session data and system logs from instrumented applications and back-end services (i.e., servers and databases) and applies statistical methods to correlate them and reduce the size of relevant log data for further analysis.

Log platforms

Monitoring systems often contain dashboards and metrics to measure the “heartbeat” of the system. In the occurrence of abnormal behavior, the operations team is able to visualize the abnormality and conduct further investigation to identify the cause. Techniques to reduce/filter the amount of log data and efficient querying play an important role to support the operations team on diagnosing problems. One consideration is, while visual aid is useful, in one extreme, it can be overwhelming to handle several charts and dashboards at once. In addition, it can be non-trivial to judge if an unknown pattern on the dashboard represents an unexpected situation. In practice, operations engineers may rely on experience and past situations to make this judgment. Papers in this subcategory focus on full-fledged platforms that aim at providing a full experience for monitoring teams.

Two studies were explicitly conducted in an industry setting, namely MELODY (Aharoni et al., 2011) at IBM and FLAP (Li et al., 2017) at Huawei Technologies. MELODY is a tool for efficient log mining that features machine learning-based anomaly detection for proactive monitoring. It was applied with ten large IBM clients, and the authors reported that MELODY was useful to reduce the excessive amount of data faced by their users. FLAP is a tool that combines state-of-the-art processing, storage, and analysis techniques. One interesting feature that was not mentioned in other studies is the use of template learning for unstructured logs. The authors also report that FLAP is in production internally at Huawei.

While an industry setting is not always accessible to the research community, publicly available datasets are useful to overcome this limitation. Balliu et al. (2015) propose BIDAL, a tool to characterize the workload of cloud infrastructures, They use log data from Google data clusters for evaluation and incorporate support to popular analysis languages and storage backends on their tool. Di et al. (2017) propose LOGAIDER, a tool that integrates log mining and visualization to analyze different types of correlation (e.g., spatial and temporal). In this study, they use log data from Mira, an IBM Blue Gene-based supercomputer for scientific computing, and reported high accuracy and precision in uncovering correlations associated with failures. Gunter et al. (2007) propose a log summarization solution for time-series data integrated with anomaly detection techniques to troubleshoot grid systems. They used a publicly available testbed and conducted controlled experiments to generate log data and anomalous events. The authors highlight the importance of being able to choose which anomaly detection technique to use, since they observed different performance depending on the anomaly under analysis.

Open-source systems for cloud infrastructure and big data can be also used as representative objects of study. Yu et al. (2016) and Neves, Machado & Pereira (2018) conduct experiments based on OpenStack and Apache Zookeeper, respectively. CLOUDSEER (Yu et al., 2016) is a solution to monitor management tasks in cloud infrastructures. The technique is based on the characterization of administrative tasks as models inferred from logs. CloudSeer reports anomalies based on model deviation and aggregates associated logs for further inspection. Finally, FALCON (Neves, Machado & Pereira, 2018) is a tool that builds space-time diagrams from log data. It features a happens-before symbolic modeling that allows obtaining ordered event scheduling from unsynchronized machines. One interesting feature highlighted by the authors is the modular design of tool for ease extension.

Discussion

Our results show that logging is an active research field that attracted not only researchers but also practitioners. We observed that most of the research effort focuses on log analysis techniques, while the other research areas are still in a early stage. In the following, we highlight open problems, gaps, and future directions per research area.

In LOGGING, several empirical studies highlight the importance of better tooling support for developers since logging is conducted in a trial-and-error manner (see subcategory “Empirical Studies”). Part of the problem is the lack of requirements for log data. When the requirements are well defined, logging frameworks can be tailored to a particular use case and it is feasible to test whether the generated log data fits the use case (see subcategory “Log Requirements”). However, when requirements are not clear, developers rely on their own experience to make log-related decisions. While static analysis is useful to anticipate potential issues in log statements (e.g., null reference in a logged variable), other logging decisions (e.g., where to log) rely on the context of source code (see subcategory “Implementation of Log Statements”). Research on this area already shows the feasibility of employing machine learning to address those context-sensitive decisions. However, it is still unknown the implications of deploying such tools to developers. Further work is necessary to address usability and operational aspects of those techniques. For instance, false positives is a reality in machine learning. There is no 100% accurate model and false positives will eventually emerge even if in a low rate. How to communicate results in a way that developers keeps engaged in a productive way is important to bridge the gap of theory and practice. This also calls for closer collaboration between academia and industry.

In LOG INFRASTRUCTURE, most of the research effort focused on parsing techniques. We observed that most papers in the “Log Parsing” subcategory address the template extraction problem as an unsupervised problem, mainly by clustering the static part of the log messages. While the analysis of system logs (e.g., web logs and other data provided that the runtime environment) was extensively explored (mostly Hadoop log data), little has been explored in the field of application logs. We believe that this is due to the lack of publicly available dataset. In addition, application logs might not have a well-defined structure and can vary significantly from structured system logs. This could undermine the feasibility of exploiting clustering techniques. One way to address the availability problem could be using log data generated from test suites in open-source projects. However, test suites might not produce comparable volume of data. Unless there is a publicly available large-scale application that could be used by the research community, we argue that the only way to explore log parsing at large-scale is in partnership with industry. Industry would highly benefit from this collaboration, as researchers would be able to explore latest techniques under a real workload environment. In addition to the exploration of application logs, there are other research opportunities for log parsing. Most papers exploit parsing for log analysis tasks. While this is an important application with its own challenges (e.g., data labeling), parsing could be also applied for efficient log compression and better data storage.

LOG ANALYSIS is the research area with the highest number of primary studies, and our study shows that the body of knowledge for data modeling and analysis is already extensive. For instance, logs can be viewed as sequences of events, count vectors, or graphs. Each representation enables the usage of different algorithms that might outperform other approaches under different circumstances. However, it remains open how different approaches compare to each other. To fulfill this gap, future research must address what trade-offs to apply and elaborate on the circumstances that make one approach more suitable than the other. A public repository on GitHub (Loghub: https://github.com/logpai/loghub) contains several datasets used in many studies in log analysis. We encourage practitioners and researchers to contribute to this collective effort. In addition, most papers frame a log analysis task as a supervised learning problem. While this is the most popular approach for machine learning, the lack of representative datasets with labeled data is an inherent barrier. Projects operating in a continuous delivery culture, where software changes at a fast pace (e.g., hourly deploys), training data might become outdated quickly and the cost of collecting and labeling new data might be prohibitive. We suggest researchers to also consider how their techniques behave in such dynamic environment. More specifically, future work could explore the use of semi-supervised and unsupervised learning to overcome the cost of creating and updating datasets.

Threats to validity

Our study maps the research landscape in logging, log infrastructure, and log analysis based on our interpretation of the 108 studies published from 1992 to 2019. In this section, we discuss possible threats to the validity of this work and possibilities for future expansions of this systematic mapping.

External validity

The main threat to the generalization of our conclusions relates to the representativeness of our dataset. Our procedure to discover relevant papers consists of querying popular digital libraries rather than looking into already known venues in Software Engineering (authors’ field of expertise). While we collected data from five different sources, it is unclear how each library indexes the entries. It is possible that we may have missed a relevant paper because none of the digital libraries reported it. Therefore, the search procedure might be unable to yield complete results. Another factor that influences the completeness of our dataset is the filtering of papers based on the venue rank (i.e., A and A* according to the CORE Rank). There are several external factors that influence the acceptance of a paper that are not necessarily related to the quality and relevance of the study. The rationale for applying the exclusion criterion by venue rank is to reduce the dataset to a manageable size using a well-defined rule. Overall, it is possible that relevant studies might be missing in our analysis.

One way to address this limitation is by analyzing the proceedings of conferences and journals on different years to identify missing entries. In our case, we have 46 after the selection process. Another approach is by applying backwards/forward snowballing after the selection process. While Google Scholar provides a “cited by” functionality that is useful for that purpose, the process still requires manual steps to query and analyze the results.

Nevertheless, while the aforementioned approaches are useful to avoid missing studies, we argue that the number of papers and venues addressed in our work is a representative sample from the research field. The absence of relevant studies do not undermine our conclusions and results since we are not studying any particular dimension of the research field in depth (e.g., whether technique “A” performs better than “B” for parsing). Furthermore, we analyze a broad corpus of high-quality studies that cover the life-cycle of log data.

Internal validity

The main threat to the internal validity relates to our classification procedure. The first author conducted the first step of the characterization procedure. Given that the entire process was mostly manual, this might introduce a bias on the subsequent analysis. To reduce its impact, the first author performed the procedure twice. Moreover, the second author revisited all the decisions made by the first author throughout the process. All diversions were discussed and settled throughout the study.

Conclusions

In this work, we show how researchers have been addressing the different challenges in the life-cycle of log data. Logging provides a rich source of data that can enable several types of analysis that is beneficial to the operations of complex systems. LOG ANALYSIS is a mature field, and we believe that part of this success is due to the availability of dataset to foster innovation. LOGGING and LOG INFRASTRUCTURE, on the other hand, are still in a early stage of development. There are several barriers that hinder innovation in those area, e.g., lack of representative data of application logs and access to developers. We believe that closing the gap between academia and industry can increase momentum and enable the future generation of tools and standards for logging.

Supplemental Information

Supplemental Information 1 Primary studies for analysis.

Each row indicates a primary study that includes the respective metadata, extracted information, and classification.

Click here for additional data file.

Additional Information and Declarations

Competing Interests

Author Contributions

Data Availability

Arie van Deursen is an Academic Editor for PeerJ Computer Science.

Jeanderson Barros Cândido is a Ph.D. student at TU Delft and is conducting his research at Adyen N.V., the industry partner of his Ph.D. program.

Jeanderson Cândido conceived and designed the experiments, performed the experiments, analyzed the data, performed the computation work, prepared figures and/or tables, authored or reviewed drafts of the paper, and approved the final draft.

Maurício Aniche conceived and designed the experiments, performed the experiments, analyzed the data, authored or reviewed drafts of the paper, and approved the final draft.

Arie van Deursen analyzed the data, authored or reviewed drafts of the paper, and approved the final draft.

The following information was supplied regarding data availability:

The raw data is available in the Supplemental File.

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
