# Peer review of "Log-based software monitoring: a systematic mapping study"

_PeerJ Computer Science, doi:10.7717/peerj-cs.489_

## Round 0.1 · original submission · Major Revisions

The reviewers provide several comments, in particular on the use of the term logging, and how you did your categorisation. They also would like to see more motivation for some of the choices you have made. Please address their comments in a new version of this paper.

Reviewer 1 ·

Basic reporting

1. The title is misleading, “Contemporary software monitoring: a systematic mapping study.” In particular, while the title is about software monitory, the survey mainly discusses the usage of software logs, including logging, log parsing, log storage, and log analysis. I suggest the authors either include papers about software monitoring that are not based on software logs or revise the title to align with the main context. For example, “Log-based Software Monitoring: A Systematic Mapping Study.”

2. In abstract, the authors mention “A holistic view of the logging research field is key to provide directions and to disseminate the state-of-the-art for technology transferring.” The term “logging” may not be accurate. “Logging” typically refers to designing logging statements in source codes and the practice that developers conduct logging is “logging practice”. The research field should be “automated log analysis”.

3. The authors mention that “In this paper, we study 108 papers … from different communities (machine learning, software engineering, and systems) ” However, it is unclear how many papers each category contains. The authors are recommended to add a table to show the related statistics.

4. Line 221 (Overview of Research Areas). The term “Log Engineering” is inaccurate and misleading. Could the authors explain why “log engineering” is an appropriate term here? I would prefer “logging”.

5. Table 3. The categorization is inaccurate. For example, “Logging practice” row mainly includes empirical studies. “Implementation of log statements” row mainly includes methodologies on what-to-log, where-to-log, and how-to-log. However, studies in the “Logging practice” row also explores these three topics. Thus, I suggest the authors to use “Empirical studies” instead of “Logging practice”.

6. In “log analysis”, the authors mention several tasks, including anomaly detection and failure prediction. However, it is unclear what are the differences between them. The definition provided in Table 3 looks similar: “Detection of abnormal behaviour” and “Anticipate abnormal behaviour”.

Experimental design

1. A line of research on what kind of log we should use should be considered, for example:
[ICSE’16] Behavioral Log Analysis with Statistical Guarantees
[ESEC/FSE’18] Using Finite-State Models for Log Differencing
[ASE’19] Statistical Log Differencing

2. I suggest the authors add a section that describes existing open-source tools and datasets for log-based software monitoring.

Validity of the findings

no comment

·

Basic reporting

This paper details a mapping study about the monitoring of modern software systems. The main result is a classification about logging research areas through the analysis of 108 papers. Then a discussion concerning the findings is reported.

The article is well written, and the authors show competence and demonstrate excellent expertise in the field. I think that behind this paper, there is a certain amount of work that must be acknowledged.
The paper is clear concerning the English language, but the authors mix English UK and English US. For example, in the Introduction Section, they use both behavior (or analyze) and behaviour (or analyse). Please check it in the overall paper.

The Introduction highlights the problem adequately, and the motivations are well justified. Relation with exiting literature is satisfactory and well discussed.
Concerning the paper structure, it does not follow the template suggested by the journal for literal review. In particular, the authors added the Discussion section as recommended by the standard journal template. However, having two separate sections allow better to highlight the results with respect to their analysis.

Experimental design

The authors used the well-known guidelines from Petersen K. et al. "Systematic Mapping Studies in Software Engineering" for having a mapping study in software engineering. However, they did not follow the guidelines. In particular, they did not define any research questions about the mapping study; they did not report a subsection containing validity threats.
Concerning the period used during the search process, it comes from 1992 to 2019 (line 213). However, in the paper's title, the authors refer to the monitoring of contemporary software, and the starting year is far from being considered modern. So please motivate why they decided to use 1992 as the starting date.
Moreover, it is unclear why they decided to update the survey through a forward snowballing and not through a repetition of the search process only for the considered period. Again, please motivate it.

Please change the sentence at line 89 to better highlight the usage of the previous guidelines described in the paper.

Validity of the findings

The Results section contains the core of the paper, and it is clear and well written. However, as already said in the study design, the results are not discussed for the hypothetical defined research questions. So please, add the RQs and then modify the results section accordingly. Also, The discussion and conclusions sections are clear and well written.

Additional comments

Please see the following from the PeerJ literal review template: Should not be used to acknowledge funders – funding will be entered online in the declarations page as a separate Funding Statement and appear on the published paper

## Minor comments
- overall paper: please hyphenate open source when used as an adjective before a noun, i.e., open source projects --> open-source projects
- line 41: and also highlight --> and also highlights
- lines 228, 229: (2)security (6)model --> add space (2) security (6) model
- line 395: located located --> located
- line 456: to large log data Later, --> to large log data. Later,
- line 488: Results indicates --> Results indicate
- line 527: analysis, Farshchi et al. (2015); Why the reference? the sentence refers to another reference i.e., Juvonen et al. (2015)
- line 594: The propose --> They propose
- line 644: Ulrich et al. (2003) shows --> Ulrich et al. (2003) show
- line 651: has been also possible --> has also been possible
- line 741: allows to obtain --> allows obtaining

---

## Round 0.2 · Minor Revisions

Please consider the remaining comments by reviewer 2. After these comments have been addressed, the paper can be accepted.

Reviewer 1 ·

Basic reporting

The review is of broad interest and is in the scope of the journal.

Experimental design

Sources are properly cited.

Validity of the findings

The goals set out in the introduction has been supported well.

Additional comments

The authors have addressed all my major concerns.

·

Basic reporting

The authors made an acknowledgeable effort towards addressing my comments, although a couple of comments could have been addressed more deeply.

Experimental design

Specifically, concerning the comment related to the usage guidelines from Petersen K. et al., the authors state that they used to only derive the classification scheme systematically by using keywording of abstracts. However, some questions naturally arise. What other standard process/guidelines did the authors use? The authors' defined process seems very similar to the Petersen K. et al., except for the definition of the research questions that are missing. Moreover, it is not clear to me the motivation concerning the research questions' absence. Furthermore, the authors used the keywording of abstracts from Petersen K. et al. paper. Why did the authors not provide the process? Therefore, I am not entirely convinced about the authors' answer.

Validity of the findings

Concerning the threats to validity section, in general, it has to contain internal validity, external validity, construct validity, and sometimes conclusion validity. Please modify the threats to validity section accordingly. A good example of threats to validity section is to the following paper: Architecting with microservices: A systematic mapping study (Paolo Di Francesco, Patricia Lago, Ivano Malavolta).

Additional comments

I think that the paper would reach a higher maturity if the above comments would be reworked. Should the editor opt for a further minor revision round in this direction, from my side, it is not needed to receive the final revision.

---

## Round 0.3 · accepted · Accept

You have carefully addressed all remaining comments, and I am happy to recommend acceptance now. Congratulations again!